# Robust Identification of White Matter Hyperintensities in Uncontrolled Settings Using Deep Learning

**Alice Schiavone** [1,2]                                         ALISCH@DI.KU.DK

**Sebastian Nørgaard Llambias** [1]                              SNL@DI.KU.DK

**Jacob Johansen** [2]                                           JJ@CEREBRIU.COM

**Silvia Ingala** [2]                                            SI@CEREBRIU.COM

**Akshay Pai** [2]                                               AP@CEREBRIU.COM

**Mads Nielsen** [1,2]                                           MADSN@DI.KU.DK

**Mostafa Mehdipour Ghazi** [1]                                  GHAZI@DI.KU.DK

[1] *Pioneer Centre for AI, Department of Computer Science, University of Copenhagen, Denmark*

[2] *Cerebriu A/S, Copenhagen, Denmark*

## Abstract

White matter hyperintensities (WMH) are associated with an increased risk of stroke, cognitive decline, and dementia. A robust, yet accurate detection of WMH can help with the prevention of more lesions from forming. The task is still challenging as the lesions are often small and irregular. Hence, we propose a robust deep learning-based method for the automatic segmentation of WMH only using fluid-attenuated inversion recovery (FLAIR) scans and MRI-specific data augmentation and compare it with state-of-the-art methods. The methods are tested on public and private data, and we show that our model is more robust to domain shift and achieves higher segmentation accuracy than the alternatives.

**Keywords:** Deep learning, domain shift, data augmentation, white matter hyperintensity.

## 1. Introduction

White matter hyperintensities (WMH) of presumed vascular origin are common findings on brain magnetic resonance imaging (MRI), typically assessed in fluid-attenuated inversion recovery (FLAIR) sequences (Pantoni, 2010). These are associated with vascular risk factors and predict an increased risk of stroke, dementia, depression, cognitive impairment, and mobility, both in cross-sectional and longitudinal studies (Wardlaw et al., 2015). Hence, segmentation and detection of WMH are crucial in the analysis of the brain.

Automatic segmentation of WMH attempts to replace the time-consuming, expensive process of manual annotation. Still, the task is challenging because WMH are often small and irregular, making the classes highly imbalanced. Besides, data availability and variability make the problem more complex due to dealing with sensitive data and differences in pathologies, anatomies, MRI scanners, and acquisition protocols.

In this study, we use 3D U-Net models (Isensee et al., 2020) and train them on FLAIR images for WMH segmentation using the common image data augmentation techniques used in (Isensee et al., 2020) and MRI-specific data augmentation proposed by Mehdipour Ghazi and Nielsen (2022). We compare the segmentation accuracy of these two methods with the state-of-the-art method (Li et al., 2018) known as the winner of the MICCAI 2017 WMH Segmentation Challenge (Kuijf et al., 2019), which benefits from both T1-weighted and FLAIR MRI scans for WMH segmentation. The obtained results show that the proposed method identifies WMH in two different datasets significantly better than the alternatives.

## 2. Methods

### 2.1. Study Data

The MICCAI 2017 WMH Segmentation Challenge training ($n = 60$) and testing ($n = 110$) datasets were used for training and testing purposes, respectively. The used datasets contain FLAIR images from five different scanners, two of which were excluded from the training set and assigned for testing. They were manually annotated as background, WMH, and other (non-WMH) pathologies. we used FLAIR images Additionally, we used an external test set of 22 FLAIR images from an in-house dataset acquired from the US and India.

Since the WMH load has a major impact on the model performance (Gaubert et al., 2023), we divided our test sets into low-load and high-load subsets concerning the volume of WMH and other abnormalities (OA). We refer to images with a load of other abnormalities greater than 1mL as WMH+OA and images without significant other abnormalities as WMH. Either group can be further divided into a "high WMH load" set if the WMH volume is greater than or equal to 10mL, or into a "low WMH load" set otherwise. The threshold values are obtained based on visual inspection.

### 2.2. Evaluation Metrics

Given the ground truth annotations and predicted segmentations, we evaluate the model performances using the overlap-based metric of Dice similarity coefficient (DSC) (Dice, 1945) and the distance-based metric of volume symmetry (VS) (Taha and Hanbury, 2015).

### 2.3. Deep Learning Models

We compare three different deep learning-based models for WMH segmentation. We use the state-of-the-art method of (Li et al., 2018), known as the winner of the MICCAI 2017 WMH Segmentation Challenge, which uses an ensemble of 2D U-Nets (Ronneberger et al., 2015). Moreover, the nnUNet (Isensee et al., 2020) is used as a segmentation framework using U-Nets with deep supervision, standard image data augmentation, and auto-configuration of network parameters. Finally, we train 3D U-Nets with MRI-specific data augmentation proposed by Mehdipour Ghazi and Nielsen (2022).

## 3. Experiments and Results

We trained the 3D models (nnU-Net and the proposed) using a combination of cross-entropy loss and Dice loss and optimized them based on the SGD method with Nesterov momentum of $\mu = 0.99$ and an initial learning rate of $\alpha = 0.01$. The training data were randomly split into training and validation sets in a 5-fold cross-validation fashion, where in each fold 80% assigned for training (48 samples) and 20% (12 samples) for validation.

Table 1: Test segmentation accuracy (mean±SD) of different models. The best results are in boldface.

| | MICCAI data | | In-house data | |
|---|---|---|---|---|
| | DSC | VS | DSC | VS |
| Challenge winner (Li et al., 2018) | .64 ± .17 | .76 ± .17 | .42 ± **.21** | .71 ± .28 |
| 3D nnUNet (Isensee et al., 2020) | .73 ± .13 | .91 ± .09 | .41 ± .27 | .63 ± .29 |
| Our proposed | **.80 ± .10** | **.92 ± .09** | **.49** ± .23 | **.72 ± .28** |

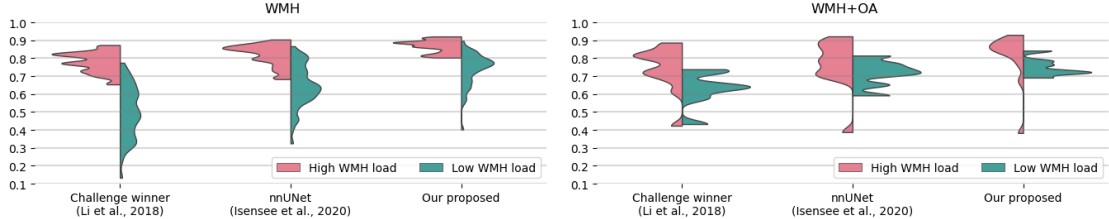

Figure 1: The obtained DSC distribution of different models on WMH segmentation of the MICCAI challenge test set with/without other abnormalities and w.r.t. the WMH load.

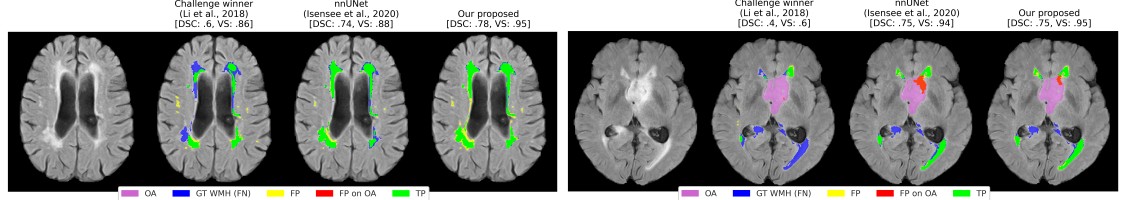

Figure 2: Two samples of the in-house data with high WMH load: a scan with WMH on the left (WMH 15.7mL), and a scan with WMH and other abnormalities on the right (WMH 19.7mL, tumor 15mL).

The obtained segmentation results are shown in Tables 1 and 2. As can be seen, by only using FLAIR images, the proposed model achieves the best WMH segmentation accuracy in both test sets compared to the alternatives. We also observe that the WMH load of other abnormalities has less impact on the accuracy of the MICCAI test set (see Fig. 1). We achieve a total DSC of 0.80 on the MICCAI test set, which is 16% better than the challenge winner. The total DSC on the in-house test set is 0.49, which is 7% better than the challenge winner. More specifically, as shown in Table 2 and Fig. 2, we obtain DSCs of 0.87 (+9%) and 0.78 (+8%), on subjects with high and low WMH load, respectively. Still, when other abnormalities are present, the accuracy drops to 0.81 (+6%) and 0.72 (+19%).

## 4. Conclusion

We proposed a robust deep learning-based method for WMH segmentation using FLAIR images. The robustness was tested using two different datasets with scans from different devices and acquisition parameters unseen to the trained models and compared the results with two state-of-the-art methods. We showed that FLAIR sequences are enough to achieve a higher WMH segmentation accuracy than the state-of-the-art, even for out-of-distribution data. However, the models cannot generalize well to the data with a low WMH load.

Table 2: Segmentation accuracy (mean±SD) of different models and loads. The best results are in boldface.

| | WMH+OA | | | | WMH | | | |
| | High WMH load | | Low WMH load | | High WMH load | | Low WMH load | |
| Methods (MICCAI test set) | DSC | VS | DSC | VS | DSC | VS | DSC | VS |
|---|---|---|---|---|---|---|---|---|
| Challenge winner (Li et al., 2018) | .75 ± **.12** | .84 ± **.13** | .63 ± .10 | .76 ± .14 | .78 ± .06 | .88 ± .08 | .50 ± .15 | .65 ± .15 |
| 3D nnUNet (Isensee et al., 2020) | .77 ± .13 | .89 ± .15 | .72 ± .07 | **.89 ± .08** | .82 ± .06 | .94 ± .05 | .65 ± .12 | .89 ± .10 |
| Our proposed | **.81** ± .14 | **.89** ± .16 | **.75 ± .05** | .88 ± .09 | **.87 ± .04** | **.95 ± .04** | **.74 ± .09** | **.90 ± .09** |
| Methods (In-house test set) | | | | | | | | |
| Challenge winner (Li et al., 2018) | .53 ± .18 | .68 ± .10 | .27 ± .18 | .56 ± .34 | .70 ± .09 | .89 ± .04 | .46 ± **.12** | **.86 ± .10** |
| 3D nnUNet (Isensee et al., 2020) | .69 ± .09 | **.89 ± .06** | .17 ± **.15** | .44 ± **.28** | .77 ± .05 | .92 ± .06 | .50 ± .17 | .69 ± .23 |
| Our proposed | **.72 ± .04** | **.89** ± .09 | **.31** ± .18 | **.61** ± .33 | **.78 ± .01** | **.96 ± .02** | **.55** ± .16 | .72 ± .24 |

## Acknowledgments

This project has received funding from Innovation Fund Denmark under grant number 1063-00014B, Lundbeck Foundation with reference number R400-2022-617, and Pioneer Centre for AI, Danish National Research Foundation, grant number P1.

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
