# OpenReview forum: "Robust Identification of White Matter Hyperintensities in Uncontrolled Settings Using Deep Learning"
_MIDL.io/2023/Short_Paper_Track — MIDL 2023 Short paper track Poster_

### Official Review · Reviewer_BwDG · 2023-04-22
**An evaluation of different U-nets for WMG segmentation, with open data and good visualisation of results**

**Rating:** 6
**Confidence:** 4

**Review:**

he paper compares different U-nets on WMH segmentation.

Pros:
- Open data is used
- Competitive baselines are used
- Dividing the evaluation into low and high load is a good idea, and clinicall relevant. Also presenting the smoothed kernel density plots this way, rather than a single scalar, gives a good understanding of performance.

Cons:
- What are the MRI-specific augmentations used? Would be good to give an idea
- The novelty is unclear - is the approach here just using different augmentations, published in the paper referenced (Mehdipour et al)? If is is the case the authors should be more explicit and argue and frame this more clearly. Why prefer this to a new architecture? Can this be combined with an nnUnet for instance?

---

### Official Review · Reviewer_pepp · 2023-04-25
**An experiemtal evaluation of two data augmentation techniques for WMH segmentation**

**Rating:** 5
**Confidence:** 4

**Review:**

This study trains a 3D U-Net model (Isensee et al., 2020) on FLAIR images for White matter hyperintensities (WMH) segmentation. It compares two existing data augmentation techniques, the one used in (Isensee et al., 2020) and another MRI-specific augmentation proposed by (Ghazi and Nielsen, 2022). The segmentation accuracies obtained by these two methods are compared to the winner of the MICCAI 2017 WMH segmentation Challenge (Li et al., 2018).

The authors report experimental results on public and in-house data, showing better performance than (Li et al., 2018), and better robustness to domain shifts.

The authors mention that the paper compares the evaluated model/augmentation combination with state-of-the-art  methods in WMH segmentation. Does the 5-year old method in (Li et al., 2018) represent the state-of-the-art in WMH segmentation? I believe a purely experimental paper, which essentially evaluates two existing augmentation techniques, should at least report more comprehensive comparisons with the state-of-the-art, including more recent methods for WMH segmentation.